# Anthropometric, Body Composition, and Morphological Lower Limb Asymmetries in Elite Soccer Players: A Prospective Cohort Study

**DOI:** 10.3390/ijerph17041140

**Published:** 2020-02-11

**Authors:** Lucia Mala, Tomas Maly, Lee Cabell, Mikulas Hank, David Bujnovsky, Frantisek Zahalka

**Affiliations:** 1Sport Research Center, Faculty of Physical Education and Sport, Charles University, Prague, Czech Republichank@ftvs.cuni.cz (M.H.); bujnovsky@ftvs.cuni.cz (D.B.); zahalka@ftvs.cuni.cz (F.Z.); 2Department of Health and Human Performance, Texas State University, San Marcos, TX 78666, USA; lcabell511@txstate.edu

**Keywords:** fat mass, fat-free mass, elite athletes, ontogenesis, talent, morphological asymmetry

## Abstract

The aim of this study was to identify and compare parameters related to anthropometry, body composition (BC), and morphological asymmetry in elite soccer players in nine age categories at the same soccer club (*n* = 355). We used a bio-impedance analyzer to measure the following indicators of BC: body height (BH); body mass (BM); relative fat-free mass (FFMr); percentage of fat mass (FM); and bilateral muscle mass differences in the lower extremities (BLD∆). Age showed a significant influence on all parameters observed (F_64,1962_ = 9.99, *p* = 0.00, λ = 14.75, η^2^_p_ = 0.25). Adolescent players (from U16 through adults) had lower FM values (<10%) compared to players in the U12–U15 categories (>10%). The highest FFMr was observed in the U18 category. Players in the U12 and U13 categories showed more homogenous values compared to older players. With increasing age, significantly higher FFMr was observed in the lower extremities. An inter-limb comparison of the lower extremities showed significant differences in the U17 category (t_27_ = 2.77, p = 0.01) and in adult players (t_68_ = 5.02, *p* = 0.00). Our results suggest that the end of height growth occurs around the age of 16 years, while weight continues to increase until 20 years. This increase is not linked to decreasing FM, nor to the FFMr, which remains stable. We found morphological asymmetries between limbs in players of the U17 category and in adult players.

## 1. Introduction

Anthropometric and body composition (BC) indicators are important factors affecting the specific attributes of today’s soccer players [1]. Together with motor coordination and physical performance measures, they have been shown to discriminate between successful and less successful youth players [2]. Generally, soccer players who are taller, heavier, more muscular, and have more active mass and less fat mass may have major advantages, especially during growth and maturation. Continuous long-term monitoring of anthropometric parameters and BC should be performed. Moreover, it is necessary to consider sensitive periods for the development of physical abilities during ontogenesis and optimize training processes and eating habits accordingly. Biological variability in anthropometric and morphological parameters appears during ontogenesis [3,4]. The variability in anthropometric indicators and BC parameters during this period can be used to identify an elite player at an adolescent age [5]. It has been reported that soccer players with increased body size dimensions have improved speed, power, and strength performance, especially during the pubertal years [6]. Conversely, several longitudinal observation studies of adolescent soccer players have shown high consistency in anthropometric measures, sprint speed, explosive leg power, isokinetic strength, and maximal aerobic speed among players [7,8], and in motor performance among players at different positions, with the exception of goalkeepers [9]. Anthropometric and morphological markers are also related to morphological asymmetries, as observed in athletes participating in asymmetric sport disciplines like tennis associated with asymmetrical changes in soft tissues [10]. When unilateral load (or a preferred limb) is involved in sport-specific movement, the limb becomes preferred by neural-motor patterns, resulting in different kinds of morphological and strength asymmetries [11].

Soccer players use both legs to kick, but they kick more often and harder with the preferred leg. Asymmetry of the lower extremities can be a consequence of unilateral kicking and greater movement repetition of the preferred limb [12]. Shortened optimal muscle length, lack of muscle flexibility, strength imbalance, and previous injuries may have severe physical consequence including increased risk of injury [13] which may negatively affect athletic performance. A study by Fousekis et al. [14] clearly describes the impact of massive amounts of repetitive unilateral movements in a soccer game or practice. As the study indicated, this imbalanced performance of limbs leads to various musculoskeletal asymmetries of lower limbs, which are associated with increased risk factors for athletic injuries [15,16]. Therefore, identification of unilateral or bilateral asymmetries of the lower limbs using various methods of investigation is crucial [17]. Muscle injury frequency caused by strength asymmetry is significantly increased for unbalanced soccer players up to 16.5%; compared to 4.1% for balanced players [16].

Besides kicking, soccer frequently involves other one-sided activities, such as different turns, tackling, or passing, which may lead to morphological and functional asymmetries [14]. However, the effect of player age on these asymmetries has not been well studied. Similarly, there is a lack of information regarding BC and morphological asymmetries in elite soccer players from the same club, at ages ranging from 12 years to adults (in nine age categories). Here, we present results from one club (at the highest national level), which has clinical practitioners in each age category (e.g. physical therapists, strength conditioning coaches). Players in this club undergo BC measurements at least twice a year, with the aim of detecting morphological asymmetries and compensating for them using a tailored training program. To the best of our knowledge, there is also a lack of studies evaluating BC of elite soccer players in the period of late maturation. As biological maturation concludes around the 17th year for most of players, fat free mass is still increasing until the age of 18. Players between the ages of 17 to 19 who meet the parameters associated with agility, aerobic power, speed, or height often play on adult teams. The novelty of this study is in the comparison of the bilateral morphology differences across all the observed age categories and especially in the periods of peak height velocity (PHV) and late maturation. At 13 to 14 years, boys experience various rapid increases in growth and bilateral morphology with soccer performance fragile to influence, as increased asymmetry is associated with risk of injury in young male soccer players [18].

The aim of this study was to identify and compare anthropometric, body composition, and morphological asymmetry parameters among nine age categories of elite soccer players from the same soccer club.

## 2. Materials and Methods

### 2.1. Study Design

A cross-sectional study design was used in this investigation. The design was explained to all participants and informed consent was collected from players, or their parents for players younger than 18 years, before assessments were performed. The study was approved by the Ethical Committee of the Faculty of Physical Education and Sport, Charles University, in Prague, Czech Republic (Nr. 101/2018). Measurements were performed according to the ethical standards of the Helsinki Declaration.

### 2.2. Subjects

The study involved 355 male soccer players, all members of the Czech cadet, junior, or senior teams. All athletes were recruited from the same club and all of them played in the highest national division in their age category. The average period of soccer training experience for each group was 13.3 years and ranged between 6.2 to 21.7 years. The research was performed in 2017 and 2018, during the pre-season player screening process. The mean ± standard deviation of the age and an overview of the typical weekly training/match frequency is shown in Table 1.

### 2.3. Data Collection

Body height (BH) was measured using a digital stadiometer (seca 242, Seca, Hamburg, Germany) and body mass (BM) was measured using a digital scale (Seca 769). Players were measured barefoot and wearing only underwear. BC measurements were taken under the same conditions, during morning hours. In the 24 h prior to the measurements, the participants did not consume any medications or pharmacological agents (including alcohol and caffeine) that could influence the results of the measurement. They were also told not to eat or drink before the measurement and to maintain good hydration and a normal routine. Furthermore, the athletes did not perform high intensity physical activity of any significant duration for the 48 h before the tests. The room temperature was kept between 20 and 24 °C to prevent undesirable changes in body water composition [19]. BC was assessed using a multi-frequency bio-impedance analyser (MC-980MA; Tanita Corporation, Tokyo, Japan), according to the manufacturer’s guidelines. Standardized conditions for bio-impedance measurements were maintained [20]. The following indicators of BC were measured: BH; BM; relative fat free mass (FFMr); percentage of fat mass (FM); and bilateral muscle mass differences in the lower extremities (BLD∆), which compared fat-free mass (FFM) in absolute values. FFMr was calculated as a normalized value of FFM to BM and BLD∆ was calculated as a percentage difference of the fat-free mass between the legs.

### 2.4. Statistical Analysis

For all variables, we calculated the following basic descriptive statistics: central tendency (mean); variability (standard deviation); and shape (skewness, kurtosis). Measurements and measures of the variability were expressed as arithmetic means and standard deviations, respectively. Parametric procedures were chosen over the normal data distribution and were verified using the Shapiro-Wilk test. Single data points were identified as outlier values if they were outside of the interval of the mean ± 2 × standard deviation.

Differences in the observed variables among the groups were assessed using a multivariate analysis of variance. We used multiple comparisons of means (Bonferroni’s post-hoc test) to compare differences in particular parameters among the groups. When the criterion of sphericity was not met as an assumption for data processing, assessed by the Mauchly’s test (χ^2^), the degrees of freedom were adjusted using the Greenhous-Geisser’s sphericity correction and statistical significance was assessed based on the degrees of freedom. We used Levene’s test of the equality of variances to verify the assumption that the error variances of the dependent variables were equal across categories.

We rejected the null hypothesis at the *p* ≤ 0.05 level. The partial eta squared coefficient (η_p_^2^), which explains the proportion of the variance contributed by a tested parameter, was used to assess effect size as follows: η_p_^2^ = 0.02 was considered a small effect, η_p_^2^ = 0.13 was considered a medium effect, and η_p_^2^ = 0.26 was considered a large effect.

Paired differences between preferred and non-preferred limbs were evaluated using a Student’s t-test for dependent variables, which was preceded by the analysis of variance based on an F-test.

The p-value, indicating the probability of a type I error (alpha), was set at 0.05 in all statistical analyses. Statistical analyses were performed using IBM^®^ SPSS^®^ v24 (Statistical Package for Social Sciences, Inc., Chicago, IL, USA; 2012).

## 3. Results

### 3.1. Body Height and Body Mass

The main factor of age showed a significant effect on BH (F_8,346_ = 64.68, p = 0.00, η_2_^p^ = 0.60) and BM (F_8,346_ = 73.66, p = 0.00, η_p_^2^ = 0.63) in male soccer players (Figure 1 and Figure 2). Post-hoc analysis showed significant differences among players in the U12–U15 categories (Table 2). BH was not significantly different among players in the U16–U19 categories (Table 2). Adult players had the highest BH values (182.5 ± 6.6 cm). Significant differences in BM were found between adult players and U12–U18 players (Table 2).

### 3.2. Fat Mass and Fat-Free Mass

Age showed a significant influence on both FM (F_8,346_ = 12.34, p = 0.00, η_2_^p^ = 0.22) and FFMr (F_8,346_ = 10.98, p = 0.00, η_2_^p^ = 0.20). FM in younger players (U12–U15) was higher than 10%, whereas older players (U16–adults) had lower FM values (< 10%, Figure 3). Multiple comparisons of means among the observed groups are presented in Table 3. The highest percentage of FFMr (91.89 ± 3.27%) was observed in the U18 category. Players in the U12 and U13 categories showed more homogenous values than older players (Figure 3 and Figure 4).

### 3.3. Segmental Comparison of Muscle Mass in Upper and Lower Extremities

Age had a significant influence on the muscle mass proportion observed in the lower extremities (Table 4). We observed a significant increase in the percentage of muscle mass with increasing age.

A comparison of the lower extremities (Table 5) revealed significant differences among limbs in the U17 category (t_27_ = 2.77, p = 0.01) and in adult players (t_68_ = 5.02, p = 0.00).

## 4. Discussion

### 4.1. Anthropometric Parameters

The primary aim of the present study was to identify and compare anthropometric, body composition, and morphological asymmetry parameters. The BH and BM measurements of the teams monitored showed values at levels previously reported for elite soccer teams [1,8,21]. Physical growth is a continuous process that occurs during the years of infancy, childhood, and puberty, and ceases when adult stature is reached. As reported by Richardson et al. [22], the rate of BM gain reaches a maximum of 20–25 kg between the 12th and 16th year (increase in FFM, decrease in FM) and from the 16th to the 20th year, it increases by 10 kg on average. The process is similar for gains in BH and BM. Consequently, the oldest boys in each age group acquire a physical advantage. In our study, the difference in BH between U12 and U15 players was approximately 7.0 cm per age group, which is consistent with normal growth at this age (approximately 7 cm every 12 months) [23,24]. The smallest and lightest players in our research group were the U12 group, with significant increases in BH in the U12–U15 categories, with the greatest difference between 13 and 14 years (increase in BH by up to 12 cm). These results confirm that this is the period of the greatest growth, which is often associated with loss of coordination of already-appropriated movements typical for this age.

The lack of significant difference in BM over two consecutive years found in the present study was consistent with data reported by Deprez et al. [8]. The study showed that during two years of monitoring, no significant differences were observed in BH or BM between players at particular performance levels (*n* = 42; pubertal soccer players, 11 and 16 years old) from two Belgian professional soccer clubs. However, over four years of monitoring, the authors reported that elite players were significantly smaller and slimmer than lower-level and average skill-level players (*p* < 0.001). After two and four years, the magnitude of the differences at baseline were reduced, although the high-performance players still showed the greatest change in BM until age 16. The optimal anthropometric profiles of successful soccer players have been investigated in several studies. Some studies have reported that soccer players with increased body size dimensions and biological maturity showed improved speed, power, and strength performance, especially during pubertal years [2,6,25,26].

### 4.2. Body composition

The results revealed a low proportion of FM and, on the contrary, a high proportion of FFMr. The main effect of age was a significant influence on FFM in the lower extremities (Table 3). Values of FFMr (87.23 ± 3.19 %) and FM (12.78 ± 3.14 %) are comparable with the study by Maly et al. [11] where they reported similar values (FFMr = 86.60±2.81%, FM = 13.40 ± 2.81%) in U15 players (*n* = 76). The optimal amount of FFM proportion may contribute to better performance (optimal physical output) as the study by Kim et al. [27] presented a significant relationship between lean body mass and VO_2_max. They concluded that active muscle mass involved during exercise is highly associated with VO_2_max in competitive rowers.

The study by Richardson et al. [22] highlights the rapid changes in FFM, FM, BH, and skeletal width, especially in males from 13 to 14 years of age. The natural chronology of body structure development, in combination with specialized elite training, supports an increase in FFM and a decrease in FM. In the present study, there were significant increases in FFMr, FM, and BH between the U13 and U14 groups (Figure 3 and Figure 4). Our results are consistent with a period of peak height velocity (PHV), which occurs in boys around 13.4 to 14.4 years [28]. The authors reported that 1 year after this period, there is a phase of peak strength velocity [28]. Degache et al. [29] reported that the greatest increase in strength occurs between the 12th and 14th years. In addition to increasing body dimension parameters (anthropometric and BC parameters) and strength, other important physical components (speed, power, agility) can also be potentiated. Reilly et al. [30] reported two stages of running speed evolution. During the first stage at the age of 8 years, players new to the game of soccer try to run as fast as they can. The second stage occurs in males from 12 to 15 years of age. In both stages, systematic training must be maintained because they are very sensitive periods of neuromuscular maturation. Consequently, this has led to the development of the term “periods of accelerated adaptation” [31], as this suggests these time periods are simply opportunities for athletes to make greater improvements in athleticism than otherwise possible.

FFM strongly contributes to strength and power performance [32,33] and is considered a major precondition for good performance in various sports and for the optimal individual performance of soccer players. Perez-Gomez et al. [33] reported significant correlation between lean body mass and peak power (r = 0.67, *p* < 0.05) and mean power (r = 0.74, *p* < 0.05) in Wingate tests. Milsom et al. [34] reported a significantly greater proportion of FFM in all segments monitored (left arm, right arm, trunk, left leg, and right leg; *p* < 0.05) in the leading team in the English Premier Soccer League, in comparison to players at the U18 level (*n* = 75). Sedano et al. [35] found a significant correlation between FFM and kicking speed in elite female soccer players and demonstrated a greater dependence between these variables in elite players than in non-elite players. FFM contributes to the production of power during high-intensity activities and provides greater absolute strength for resistance to high dynamic and static loads.

### 4.3. Segmental Comparison of Upper and Lower Extremity Muscle Mass

Morphological asymmetry, defined as the difference between the right and left part of the body, is most likely a consequence of specific, mostly asymmetrical movements, without adequate compensation.

We found no significant differences in the morphology of the lower extremities in the U12–U16 categories (Table 5). We assumed morphological asymmetries of the lower limbs, particularly in the group of the older players, was due to a higher volume of training sessions and higher specificity of movements (rate of unilateral motor performance execution). This hypothesis was only confirmed in some age categorie

The asymmetries evaluated may have been caused by functional adaptations, when the muscle groups successfully adapt to the asymmetrical loading demands of soccer in order to decrease the excessive strain on some tissues. Some level of asymmetry may be caused by a natural effect of lateral dominance due to the unilateral nature of the sports discipline [11]. It has been reported that the combination of cyclic and acyclic movements at irregular intervals and preferential use of one extremity may lead to morphological and strength asymmetries in soccer players [14]. These asymmetries may result in large changes in the myodynamic characteristics of the muscle, particularly in the dominant leg. When the execution leg is working much more than the supporting leg, an imbalance can occur between muscle groups on the left and right side of the body [36]. Thus, the results indicate a critical period at 15 to 16 years of age for revealing the consequences of morphological asymmetries in elite players. These asymmetries may be related to functional strength asymmetries in soccer players, which has already been reported [37]. Read et al. [18] evaluated significantly higher single-leg 75% hop and stick landing performance force asymmetries in different stages of maturation in youth elite soccer players. Provided that morphological asymmetries are the consequence; their real cause occurs much earlier. They may be associated with the unilateral preference of a limb, which is not properly compensated by an appropriate number of repetitions for the second limb or through a suitable compensation plan.

In the terms of age and the occurrence of morphological asymmetries, Tsolakis et al. [38] reported that in sports where limb preference is remarkably accentuated (e.g., fencing), morphological asymmetries tend to occur much earlier and their occurrence is dependent on age. Arm asymmetries were found at the age of 10–13 years, whereas leg asymmetries were observed among 14- to 17-year-old athletes. Unilateral load is required in certain sport specializations (e.g., tennis, fencing, ice-hockey, javelin throw). Asymmetrical loading in the long term may lead to asymmetry in professional tennis players due to existence of approximately 20% more bone mineral content and muscle mass in the dominant arm [39,40]. It has been reported that long-term tennis playing during childhood is associate with reduced subcutaneous adipose volume, but no significant differences between the arms of 10- to 12-year-old players [41]. On the other hand, non-significant differences of asymmetry (muscle volume) in iliopsoas and gluteal muscles between dominant vs. non-dominant legs were observed in soccer players [42].

Proportional and harmonious development of anthropometric, morphological, and functional determinants of sports performance is necessary for the proper development of talented athletes and for achieving maximum physical performance. For proper execution of a specific technique (e.g., kicking, passing), length of running stride, optimal coordination during rapid changes in the direction of movement, and other motor tasks, optimal range of joint mobility is important. Muscle imbalance and shortening may limit the range of joint mobility, and thus become a limiting factor for the proper execution of a soccer player’s technique. Asymmetry may have multiple consequences. Therefore, it is very important to act and compensate for this pathology. In sport, there is a need to seek certain solutions to achieve optimal results, especially in young athletes. Moreover, it is necessary to investigate the association between morphological and function asymmetries in youth and adult elite soccer players.

### 4.4. Limitations

Despite the results presented herein, we are aware the present study has several limitations. The research was only performed with elite players from one professional club, where all categories had a physiotherapist, fitness coach, and massage therapist (at least one). It is possible that the results of a lower level or different club may be different. Furthermore, we did not divide players based on their field positions (because of the lack of players for this analysis). The next limitation is that we only controlled chronological age. In future studies, it will be necessary to control for biological maturation or use a bio-banding approach [43]. A primary study limitation is the lack of a gold-standard tool (DEXA), because of the high number of unsigned informed consents of participants’ parents for DEXA measurement. A further limitation of this study is that we did not consider sexual maturation (Tanner stage) because we compared players according their training groups in their teams (chronological age, performance, competitive level) and state of hydration. Moreover, generalizability of these data is limited, since all players in the study were males. Future studies that include females, as well as soccer players with amateur and recreational status, are warranted.

## 5. Conclusions

Our data showed that the knowledge and long-term monitoring of BC factors, such as FM, FFMr, and the incidence of morphological asymmetries of players could give a coaching staff valuable information about the players, with the aim of improving their performance. Our study confirms the end of height growth around the age of 16 years in male soccer players, while showing that weight continues to increase until 20 years. This increase is not linked to decreasing fat mass, nor to the fat free mass, which remains stable. It is, rather, an increase in bone mass. Tensions are therefore maximum at 16 years (end of growth in size) while the bone is still fragile. Asymmetry effects stemming from different segmental muscle mass proportions between the preferred and non-preferred limbs and may increase a player’s risk of injury. The main finding of our study was the absence of morphological asymmetry in youth players (U12–U16 categories); however, regular monitoring should occur in these players and target-specific exercises implemented if asymmetries are found. Our study revealed morphological asymmetries between limbs in players in the U17 category and in adult players. More attention should be paid to players in these categories because of these predictors of injury risk. Asymmetries should be systematically monitored and compensated with specific exercises. The current study data are likely feasible to be used to construct an effective morphology status (youth talent identification and development) based on physical maturation/developmental stages and soccer positions. Also, it should be beneficial for clinical practitioners such as nutritionists or physical therapists in terms of morphology asymmetries identification and compensation by properly tailored strength and conditioning programs.

## Figures and Tables

**Figure 1 ijerph-17-01140-f001:**
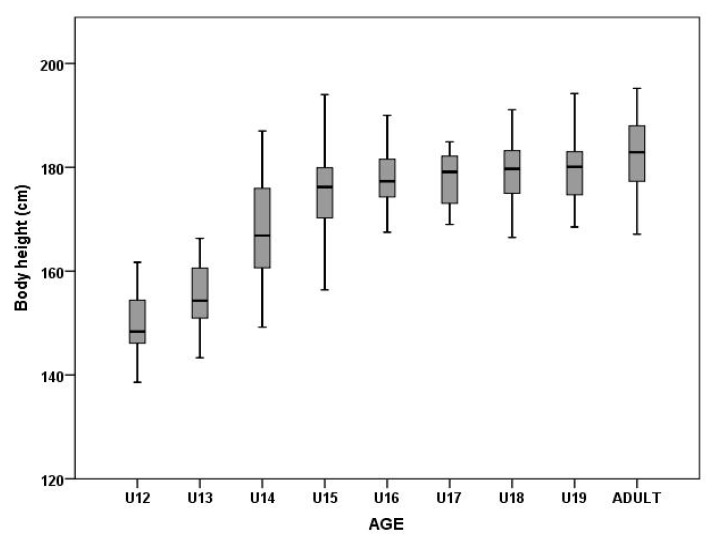
Differences in body height among different age groups.

**Figure 2 ijerph-17-01140-f002:**
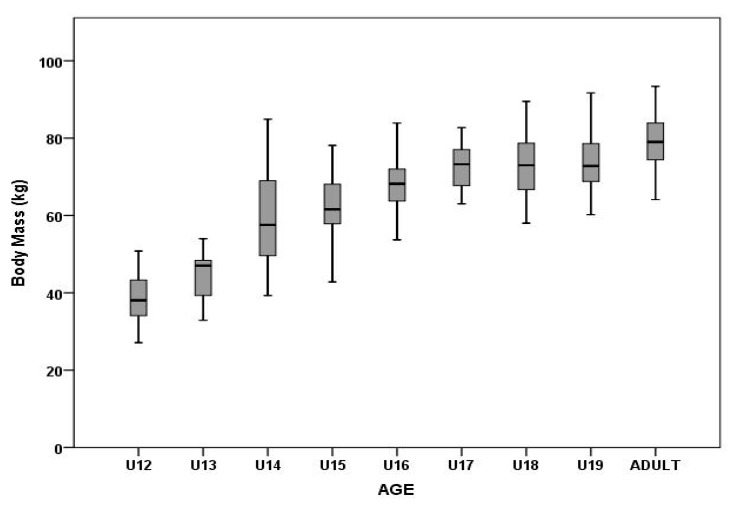
Differences in body mass among different age groups.

**Figure 3 ijerph-17-01140-f003:**
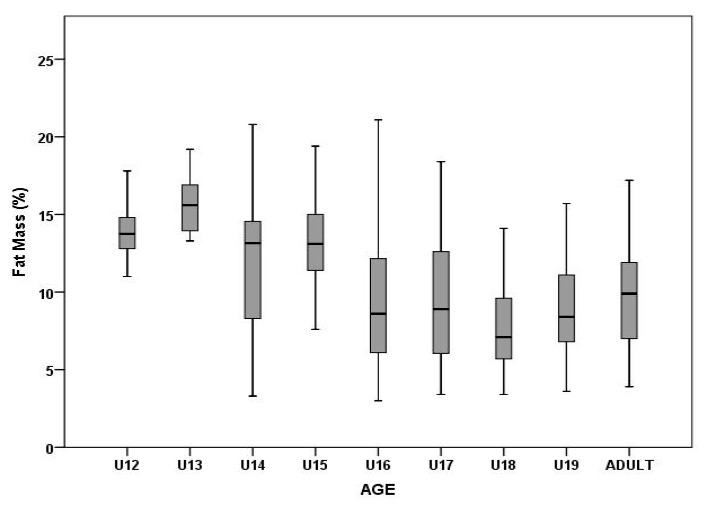
Differences in fat mass among different age groups.

**Figure 4 ijerph-17-01140-f004:**
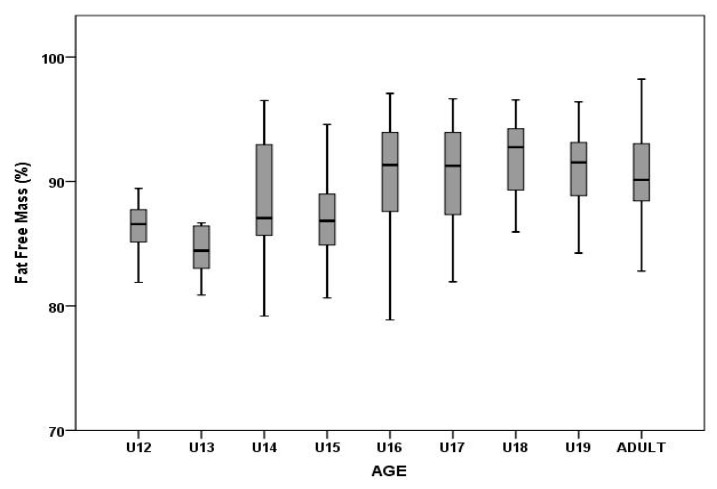
Differences in fat-free mass among different age groups.

**Table 1 ijerph-17-01140-t001:** Chronological age and overview of weekly training/match frequency for selected groups.

Team	*n*	Age (years)	Field-BasedTraining	ResistanceTraining	Match
Adult	69	23.1 ± 2.6	6–7 × (60–90 min)	1 × 45 min	2 × 45 min
U19	33	18.8 ± 0.6	5–6 × (60–90 min)	1 × 45 min	2 × 45 min
U18	41	17.6 ± 0.4	5–6 × (60–90 min)	1 × 45 min	2 × 45 min
U17	28	16.6 ± 0.5	5–6 × (60–90 min)	1 × 30 min	2 × 40 min
U16	71	15.6 ± 0.6	5–6 × (60–90 min)	1 × 30 min	2 × 40 min
U15	47	15.7 ± 0.3	5 × (60–90 min)		2 × 35 min
U14	32	13.7 ± 0.4	5 × (60–90 min)		2 × 35 min
U13	16	12.5 ± 0.6	4 × (60–120 min)		3 × 30 min
U12	18	11.8 ± 0.2	4 × (60–120 min)		2 × 35 min

**Table 2 ijerph-17-01140-t002:** Bonferroni post-hoc analysis of body height and body mass (white background, body height; grey background, body mass).

	U12	U13	U14	U15	U16	U17	U18	U19	ADULT
U12		n.s.	U12 < U14	U12 < U15	U12 < U16	U12 < U17	U12 < U18	U12 < U19	U12 < AD
U13	n.s.		U13 < U14	U13 < U15	U13 < U16	U13 < U17	U13 < U18	U13 < U19	U13 < AD
U14	U12 < U14	U13 < U14		U14 < U15	U14 < U16	U14 < U17	U14 < U18	U14 < U19	U14 < AD
U15	U12 < U15	U13 < U15	n.s.		n.s.	n.s.	n.s.	n.s.	U15 < AD
U16	U12 < U16	U13 < U16	U14 < U16	U15 < 16		n.s.	n.s.	n.s.	U16 < AD
U17	U12 < U17	U13 < U17	U14 < U17	U15 < 17	n.s.		n.s.	n.s.	n.s.
U18	U12 < U18	U13 < U18	U14 < U18	U15 < 18	n.s.	n.s.		n.s.	n.s.
U19	U12 < U19	U13 < U19	U14 < U19	U15 < 19	U16 < 19	n.s.	n.s.		n.s.
ADULT	U12 < AD	U13 < AD	U14 < AD	U15 < AD	U16 < AD	U17 < AD	U18 < AD	n.s.	

Legend: <: significant lower value between compared pairs; n.s.: non-significant differences between compared pairs.

**Table 3 ijerph-17-01140-t003:** Bonferroni post-hoc analysis for fat mass and fat-free mass (white background, fat mass; grey background, fat free mass).

	U12	U13	U14	U15	U16	U17	U18	U19	ADULT
U12		n.s.	n.s.	n.s.	U12 > U16	U12 > U17	U12 > U18	U12 > U19	U12 > AD
U13	n.s.		U13 > U14	n.s.	U13 > U16	U13 > U17	U13 > U18	U13 > U19	U13 > AD
U14	n.s.	U13 < U14		n.s.	n.s.	n.s.	U14 > U18	n.s.	n.s.
U15	n.s.	n.s.	n.s.		U15 > U16	U15 > U17	U15 > U18	U15 > U19	U15 > AD
U16	U12 < U16	U13 < U16	n.s.	U15 < 16		n.s.	n.s.	n.s.	n.s.
U17	U12 < U17	U13 < U17	n.s.	U15 < 17	n.s.		n.s.	n.s.	n.s.
U18	U12 < U18	U13 < U18	U14 < U18	U15 < 18	n.s.	n.s.		n.s.	n.s.
U19	U12 < U19	U13 < U19	n.s.	U15 < 19	n.s.	n.s.	n.s.		n.s.
ADULT	U12 < AD	U13 < AD	n.s.	U15 < AD	n.s.	n.s.	n.s.	n.s.	

Legend: <: significant lower value between compared pairs; >: significant higher value between compared pairs; n.s.: non-significant differences between compared pairs.

**Table 4 ijerph-17-01140-t004:** The effect of age on inter-limb comparison.

Compared variables	Type III Sum of Squares	Mean Square	F	Sig.	Partial Eta Squared
Muscle Mass PL (%)	142.43	17.80	21.36	0.00	0.33
Muscle Mass NL (%)	206.98	25.87	38.93	0.00	0.47

Legend: PL: preferred leg; NL: non-preferred leg.

**Table 5 ijerph-17-01140-t005:** Paired differences (%) between preferred and non-preferred limbs of soccer players.

Age Category	Preferred Limb		Non-Preferred Limb		*t*
	Mean	SD	Mean	SD	
U12	13.79	0.76	13.61	0.83	−0.91
U13	14.02	1.33	13.89	1.35	0.02
U14	14.61	1.16	14.49	1.13	0.37
U15	14.90	1.02	14.71	1.00	−0.38
U16	15.07	0.85	14.55	0.79	1.37
U17	15.26	0.89	14.73	0.83	2.77 *
U18	15.50	0.88	15.12	0.47	−0.39
U19	15.74	0.78	15.32	0.59	0.33
Adults	16.09	0.77	15.91	0.66	5.02 **

Legend: *: significant differences p < 0.05; **: significant differences p < 0.01.

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
