# Peer review of "Anthropometric, Body Composition, and Morphological Lower Limb Asymmetries in Elite Soccer Players: A Prospective Cohort Study"

_ijerph, 2020, doi:10.3390/ijerph17041140_

Round 1
Reviewer 1 Report
This study reports cross-sectional data of height, weight and body composition by BIA, of elite soccer players from the Cech Republic. The main reason to carry out a study to report novel findings. The main question I have what is adding this study to the current knowledge. Maybe, there is no other study examining the asymmetries between kicking leg, and the contralateral leg is soccer players of different ages from U12 to adulthood. This needs to be emphasized. The main limitation of the manuscript, however, is the poor quality of the references used in many instances. The authors have ignored the most relevant studies where using state-of-the-art methods, muscle asymmetries have been assessed in young soccer players (I provide some, but there are some more). The main limitation of the study is the use of BA to determine body composition and segmental body composition. BIA BC equations are population-specific and are different for adults and children. The equation used in this study is critical and should be reported based on sound references.
Major concerns
1) Referencing. The manuscript needs to be cured of irrelevant references (remove all references to non-indexed journals) o make sure that the paper is outstanding, written by known researchers or has a specific criterion of quality. Please, avoid attributing statements that do not correspond to the experimental findings of the cited paper. Select the best article to defend your statement. One strategy is to check for the citations that the references or the “category” of the research team. Give priority to cite first the authors who mage pioneer contributions, then the confirming studies.
2) Nobody knows whether asymmetries are disadvantageous or the opposed for elite athletes. Is this demonstrated in reference 12? Or just a hypothesis?
3) The revision of the literature is discouraging. Search for the many papers by Sanchis-Moysi et al. He has studied asymmetries with DXA and MRI in tennis players, tennis players compared with soccer players including young (pre-pubertal), older kids and professionals. Thus, the introduction must be rewritten with appropriate inclusion of the previous studies.
4) Add information regarding sexual maturation (Tanner stage) if available. If not comment of this briefly as a limitation.
5) Report on a new table 1. Age, BM, training volume and performance characteristics of each age group. Information is needed on the physical tests (surely these were performed).
6) AVOID the term fluid distribution: you did not measure fluid distribution, BIA measures electrical impedance. Form the latter; lean mass can be estimated with a series of assumptions.
7) Report the BIA equations used to convert the resistance data into FFM and FM.
L251-257. DELETE, this was mentioned in the results.
L274 “acceted” accentuated?
L277 Refer to Sanchis-Moysi papers, who has analyzed asymmetries with MRI in upper and lower limbs, and also in the trunk muscles.
ADD to the limitation section: the small number of subjects per category, as well as the use of electrical bioimpedance for body composition analysis.
Minor concerns:
Delete: “quality” after BC. Otherwise, you would need to define what its “quality” (more muscle mass? According to wat studies). Better, just leave “BC”.
Delete reference “5”.
L39-40 should also be deleted. You mention the relevant information in the following lines.
L48. Remove reference 10. This reference refers to tennis players.
L40 In general, use references to indexed journals (WoS). Sure, you know many more studies showing the effects of asymmetrical loading in sports. Tennis, by large, has been more studied.
L51-53. Rewrite. Soccer players used both legs to kick, but they kick harder and more often with the preferred leg.
L53-54. Reference 13 did not show that shortened muscles increase the risk of injury.
L54. Reference 12 is not needed here. You do not need a reference here; this is common sense.
L75 Delete this line and reference 14. You only need this reference when submitting to IJSM. This information is redundant to the fact that you followed the Declaration of Helsinki; this is enough.
L78. Czech cadet: Where all the players members of the national teams. If this was the case, please be clear. Then in the lines above is not correct to say “from the same soccer club” (L66).
L88 What did you mean by recovery procedures: these should be explained if based on exercises.
L88-92. More information needed. Where the subjects fasted? How was height measured? How was weight measured (clothes on/off)? Reported whether the state of hydration was determined or not (likely not). Was physical activity controlled for the day before measurements? How were FFMr and BLDΔ calculated?
L103 What did you do with the outliers?
L116: Please make it clear that you used a two-tailed t-test. If not redo the statistical analysis using a two-tailed t-test.
L117-120. Remove the Cohen’s coefficient of effect size, “d”. This does not add any value to the results section, make it more challenging to read, and the attributed meaning to the “d” values are 100% subjective. You just need to assess the physiological implication of the asymmetry detected (if any). Remove reference 16, which may be suitable for a psychologist, but in sports performance, neither biology (we use higher standards).
Results
You do not need to provide that much information about the statistical test (simplify this: for example, report the p-value and statistical power if you want too).
Delete L127-127. The same information is given below.
L132 Replace “showed no significant differences among” by “was not significantly different among”. Report the exact p-value for non-significant effects (not P>0.05).
Report height and weight with only ONE decimal.
L134. Delete “however, they were not significantly taller than the U19 (180.04 ± 6.51 cm), U18 (179.41 ± 6.25 cm), or U17 players (177.59 ± 4.98 cm).” and “However, no differences in BM were observed between U12 vs U13, U14 vs U15, U16 vs U17, U18 and U19 vs U17, U18, adult players” this information is given in Figures 1 and 2.
Delete table 1. This table is unnecessary because much better information is given in Fig 1 and 2. Add “*” just to indicate when a group is different from adults. The rest is irrelevant. The same applies to figures 3 and 4.
For ALL figures in the legend add the number of subject “n” for each age category.
Delete Table 2.
Table 3. This table is not understandable. Report the muscle mass in kg with two decimals for this variable. It is not clear what is compared with what. How many subjects were used to make this comparison? Mean square, F should be removed.
L165-166. If this statement was not statistically significant in the t-test, please DELETE.
DELETE table 4. This information is already given in Fig 6.
In ALL places, use “.” To indicate decimal position not “,” (this applies to text, tables and figures).
Figures 6 is likely in Figure 5 (otherwise the is a figure lacking). Either in the legend or on top of each bar add “n” for each age category.
L179 add “during the year” or add “in January” before advantage.
L182. According to which growth tables? And reference?
L184-185. Delete “confirms that this is the period of greatest 184 growth, which is often associated with incoordination of already appropriated movements typical for this age” Coordination was not measured in this study.
L186. Delete. You did not measure this in the present paper.
L187-189. This is irrelevant and can be deleted (is just a repetition of the Results section).
L190.193 You only have cross-sectional data: this does not allow to infer anything regarding growth.
Delete lines 202-205. This is already reported in the results section.
Delete 205-207. This is irrelevant.
FFMr and FM: report these with only ONE decimal.
L212. A better reference than 22 is needed. Malina already showed that the primary determinant of VO2max in children is lean body mass.
L229. Add a reference to PMID: 18340455, this was one of the first studies clearly showing the importance of muscle mass for power output and sprint performance, after reference 27.
L1237-239 delete.
L243 replace asymmetrical by asymmetric. Delete “neurological”
L245-246. DELETE, or expressed it right. There may be differences in bone architecture between limbs loaded asymmetrically, particularly if loading starts before puberty: this should be mentioned (Kannus papers on this particular with tennis players were the first to report this). Reference 30 must be deleted.
Some references studying muscle asymmetries in soccer players:
Look in Pubmed for:
asymmetry soccer magnetic resonance
PMID: 29318166
PMID: 29161759
PMID: 28787266
PMID: 21829539
Author Response
Thank you for all suggestions, advices and comments. Each of them are very useful and should improve our manuscript. Base on consideration all for reviewers reports and time pressure (10 days) we did in manuscript several corrections (or justification), see attached file please.

Reviewer 2 Report
This interesting study confirms the end of height growth around 16 years in men, while the weight continues to increase until 20 years. This increase is not linked to the decreasing fat mass, nor to the fat free mass which remains stable, it is therefore an increase in bone mass. Tensions are therefore maximum at 16 years (end of growth in size) while the bone is still fragile.
In the absence of a control group, it would be interesting to indicate the theoretical curve as a function of age.
In the title a prospective study design is mentioned while in the text, the study design specifies that it is a cross sectional study. The data was collected between 2017 and 2018, but I imagine the data was collected only once for the 265 subjects.
If the information is collected or measures the variables only one time, it is a cross sectional design. If the variables are measured two or more times it is a longitudinal prospective design.
Author Response
Thank you for all suggestions, advices and comments. Each of them are very useful and should improve our manuscript. Base on consideration all for reviewers reports and time pressure (10 days) we did in manuscript following corrections (or justification).
Because we obtained reports from 4 reviewers and some comments are same/similar we are sending you all responses.
If you want to check just your comments/responses see page 16 in attached document please.
Once again, thank you for your time, energy and important inputs.
Best regrads,
authors.

Reviewer 3 Report
The background should elaborate more about lower limbs asymmetries. What is known in the literature. In addition, persuade the readers what merit you have in your study. What do you contribute that was not known before.
In the method section I did not find how many participants were there in each of the 9 age groups.
In the Results section - why Figures and not Tables?!!
Tables 1 and 2 - with the signs < or > - where is the significance information?
Overall, very strange and unusual way to present the Results.
Limitations are detailed, but these are strong limitations that decreases the soundness of this study.
Author Response
Thank you for all suggestions, advices and comments. Each of them are very useful and should improve our manuscript. Base on consideration all for reviewers reports and time pressure (10 days) we did in manuscript following corrections (or justification).
Because we obtained reports from 4 reviewers and some comments are same/similar we are sending you all responses.
If you want to check just your comments/responses see pages 17-19 in attached document please.
Once again, thank you for your time, energy and important inputs.
Best regrads,
authors.

Reviewer 4 Report
Thank you for the opportunity to review the present manuscript. The authors have decided to investigate a timely and relevant topic related to anthropometry, body composition (BC), and morphological asymmetry in soccer players. However, there are several methodological concerns to this work that require clarification and preclude this paper to be published in its current form. In addition, there are several typos throughout the MS. Having said that, a major revision of the present MS is warranted.
Please see detailed comments below, line by line:
Introduction:
Please develop a study rationale, a study hypothesis, why is your research new and noteworthy?
Methods:
Any data on the intra-assay reliability of the bioimpedance measurements?
This is a critical factor of your study, since you did not use a gold-standard assessment tools, such as ultrasound imaging or DEX-a. What we do know from previous research is that the bioimpedance measurements are not consistent across all circumstances.
Please remove row lines from Tables 2-4; they will read better than.
Discussion section:
An opening line of the discussion section should be related to the primary findings of your paper. I don’t see that in your MS. Therefore, please re-write this portion of the MS.
Please start this paragraph with….”The primary aim of the present study was to”…… and from there please move to key findings observed.
Line 203 – please explain, what does BC quality mean? Or represent?
Study limitations: I would rather say that a primary study limitation is the lack of a gold-standard tools application in the present MS.
Some practical recommendation for coaches/ runners would be a viable addition to the MS at this point.
Author Response
Thank you for all suggestions, advices and comments. Each of them are very useful and should improve our manuscript. Base on consideration all for reviewers reports and time pressure (10 days) we did in manuscript following corrections (or justification).
Because we obtained reports from 4 reviewers and some comments are same/similar we are sending you all responses.
If you want to check just your comments/responses see pages 2é-21 in attached document please.
Once again, thank you for your time, energy and important inputs.
Best regrads,
authors

Round 2
Reviewer 1 Report
Some aspects of the manuscript have been improved. Unfortunately, the authors disregarded several of the suggestions made to improve their manuscript. The fact that they persist in keeping two decimals for height when the precision of measurement is no more than 1 mm, is not justified. The insistence in keeping irrelevant, repetitive and busy tables is unacceptable. That other authors have used a tool to measure something and published a paper does not mean the tool is appropriate. The equation used to estimate the asymmetries are not reported. The conclusions are not supported by the study, a cross-sectional study does not allow to infer anything meaningful regarding growth. Overall the writing style falls below the minimal standard of a scientific manuscript, apart from the numerous grammatical mistakes still present in the manuscript.
Reviewer 2 Report
I suggest accepting this article. The bibliography has been completed. The text has been made more interesting for the reader, particularly in terms of discussion. It provides an anthropometric reference useful in sports, but also in general medicine.
Reviewer 4 Report
Thank you for following my instructions. The authors have done a good job.
Congratulations.